# Intervention for Better Knee Alignment during Jump Landing: Is There an Effect of Internally vs. Externally Focused Instructions?

**DOI:** 10.3390/ijerph191710763

**Published:** 2022-08-29

**Authors:** Inge Werner, Monika Peer-Kratzer, Maurice Mohr, Steven van-Andel, Peter Federolf

**Affiliations:** 1Department of Sport Science, University of Innsbruck, 6020 Innsbruck, Austria; 2Praxisgemeinschaft MOVIA MED, 6020 Innsbruck, Austria

**Keywords:** focus of attention, instruction, knee alignment, constrained action, freezing, movement technique, principal component analysis (PCA)

## Abstract

Externally focused attention is known to induce superior results in the movement outcome, whereas focusing attention on the moving body (internal focus) causes conscious control and constrains action. The study investigated effects on knee trajectory and whole-body movement complexity when addressing knee alignment using externally (EF) vs. internally (IF) focused instructions. Young ski racers, *n* = 24 (12 male), performed landings with subsequent jumps to submaximal height. Movements were tracked and analyzed during the ground contact phase. Sets of jumps were executed without instruction (CON), followed by EF and IF instructions on knee alignment in a random order. Medial–lateral displacement of the knee in landing quantified task achievement, and whole-body principal component analysis was used to compute movement complexity. Knee alignment instructions led to a significantly lower medial knee displacement compared to CON (*p* = 0.001, η_p_^2^ = 0.35). EF vs. IF did not reach significance. EF, as well as IF instructions increased the prominence of the first movement pattern (*p* = 0.01, η_p_^2^ = 0.22) with a reduction of higher-order patterns (*p* = 0.002, W = 0.11), suggesting a strategy of freezing degrees of freedom. Both instructions addressing the movement form positively influenced knee displacement during landing, and both led to a freezing strategy, simplifying whole-body coordination.

## 1. Introduction

In sports coaching, instructions on movement execution aim to enhance performance and motor learning. These instructions may address the movement outcome (e.g., jumping distance for the standing long jump) or the movement form (e.g., usage of the arm swing within the movement technique). A long tradition of research exists comparing the effectiveness of different kinds of attentional focus prompted by instructions [1,2]. An external focus of attention is oriented towards the movement effect in the environment, whereas an internal focus of attention aims at the moving body. Various studies investigated the effect of attentional focus in different types of movements, such as balance tasks, standing long jump, or counter movement jump [3,4,5], as well as in different sports, such as darts, golf, or surfing [6,7,8,9]. Most of these studies confirm that an external focus of attention leads to enhanced movement performance and learning (for a review, see [2]), but the explanation of this phenomenon and methodical questions are still being discussed [10,11,12].

Whereas many experiments focused on movement outcome, only a few studies concerned instructions on movement form [9,13,14]. In addition, studies observing focus effects on changes in movement coordination itself are rare [15,16,17]. Instructions on movement form are of importance, not only to improve aspects of the movement technique, but also to avoid harm through overloading due to inaccurate movement execution. For instance, injury prevention programs for landing and cutting maneuvers use movement instructions to achieve better alignment of knee bending trajectories in the frontal plane [18,19,20]. Welling et al., studied the effects of different attentional foci on the jump landing technique using a standardized task to identify individuals at risk for future ACL injury [21,22]. Both the external and internal focus were exclusively related to the jump outcome and not the movement form [21]. Whether and how whole-body movement is changing when utilizing external vs. internal focus instructions on knee alignment in particular was not explored and is still unanswered. Internally focused instructions on the moving body have been said to provoke conscious control by the moving person, which can impair motor performance and learning, especially in late stages of the learning process [23].

The current study applies a principal component analysis (PCA) to quantitatively characterize these changes in coordination of movements and movement patterns [24]. PCA detects correlated movement trajectories of body landmarks, projecting the whole-body movement into independent sets of movement components [25], so-called principal movements (PMs) [26]. In most studies concerning movement techniques, a small number (e.g., 3 to 8) of principal movements are analyzed since they typically already describe most (90% and above) of the total variance [24,25,27]. The number of PMs needed to describe most of the variance can be interpreted as a measure of movement complexity, the number of independent movement patterns representing the whole movement [28,29]. As a consequence, the portion of each principal movement component within the whole movement execution could be a tool to objectively detect movement complexity changes [27]. In their study of single-leg landings, Nordin and Dufek showed that higher task demands (landing from higher heights and/or additional load) yielded a reduction in the number of principal components, expressing fewer available motor patterns [28]. Consequently, the reduction in the number of PMs needed to represent the movement, or higher portions for dominant PMs within the movement coordination, could deliver a measure for constrained action, being a consequence of internally focused instructions [1].

The aim of this study is to compare the effect of an external vs. an internal focus of attention in instructions on knee alignment when performing a land-then-jump task. Two hypotheses were explored: (i) An external focus of attention (the knee movement effect in the environment) enhances knee alignment during the automatized skill of landing-then-jumping without changing the complexity of movement patterns, when compared to landing-then-jumping with internally focused instruction or no instruction. (ii) An internal focus of attention (focusing on the knee trajectory itself) causes a more tightly controlled movement, which will manifest in a less complex structure, detectable through fewer PMs representing most of the movement variance, when compared to landing and jumping with externally focused instruction or no instruction.

## 2. Materials and Methods

### 2.1. Participants

Since knee alignment plays an important role in injury prevention in ski racing, high-level skiers were recruited. Young ski racers (*n* = 24; 15 to 16 ys of age; 12 male), well trained and free of knee injuries, volunteered in our study. Current pain, referring to the ankle joint and knee joint, as well as lower back pain in particular, served as exclusion criteria. The test protocol met the Declaration of Helsinki and was approved by the Board of Ethical Questions in Science of the University of Innsbruck (Certificate 35/2017). The study was conducted in the preseason period with high engagement of all participants. Athletes were familiar with intense training and the conversion of movement instructions. They were not aware (biased) of effects of attentional focus in given instructions, but were aware of the knee alignment problem for injury prevention in training and competition.

### 2.2. Experimental Design

After a short individual warm-up, every participant performed three sets of five jumps from a 40 cm-high platform, positioned close to a force plate, with the goal of a smooth landing, fluently followed by a vertical jump to submaximal height (landing-then-jumping task). After one trial to familiarize them with the task and equipment, a first set of five jumps was conducted without instructions concerning knee alignment. The next two blocks of five trials occurred under specific instructions (external or internal focus of attention) in a randomized order to avoid sequence-specific adaptations to the instructions.

A reflective marker was mounted on the patella of each knee and on the base joint of the second toe of each foot, respectively, being part of a 41-marker set (see below). For clear specification of these markers, they were called the knee-marker and foot-marker, defining outside positions of the body (marker height of about 1 cm). The wording of the given instructions was designed to meet the scientific claim regarding focus instructions that (i) the region of focus is meaningful to the movement and (ii) the wording of instructions differs only in small parts of the text to ensure the potential of the outcome interpretation [2,30]. The internal focus instruction (“when landing and jumping, focus on keeping the middle of your knee in line with the middle of your foot”) and the external focus instruction (“when landing and jumping, focus on keeping the knee-marker in line with the foot-marker”) were presented in written form, read by the tester, and repeated before the third trial of the set. Before every trial, a short version of the focus instruction was repeated (“focus on the middle of the knee”, “focus on the knee-marker”, respectively). In this study, the target of the external focus instruction was clearly situated outside of the body and not confused by additional visual cues for knee navigation in the environment, albeit the distance to the body was small. The strength of these instructions lies in the subtle difference between instructions with the focus moving from inside to outside of the body, without providing additional information. An observed effect of external vs. internal instruction would allow strong arguments for using either an external or internal focus when the goal is to improve the movement technique.

In previous studies, it was shown that athletes preferring a certain focus behave differently in focus conditions depending on familiarity [31]. To control for preference effects, participants were asked at the end of the experiment which of the trial blocks felt easier for them with respect to controlling the knee movement throughout the execution of the task. Participants who considered the EF easier were assigned to the external-focus-preferring group (EFpref, 13 athletes); others preferring the IF, or who could not decide, were assigned to the internal-focus-or-both group (IFboth, 6 and 5 participants, respectively). This information was used to check for preference effects on the results. Every participant was tested alone and was asked not to talk to other participants about the tests and instructions to minimize bias.

### 2.3. Data Collection and Data Analysis

The athletes were tracked with a Vicon system using 8 cameras (Vicon motion systems, Oxford, UK). They were equipped with 41 markers according to the plug-in-gait marker placement [32]—with 4 head markers, cervical spine C7, thoracic spine T10, clavicular and sternum, right scapula; pairwise left and right: shoulder, upper arm, elbow, wrist inside and outside, finger, pelvic front and back, thigh, knee, shank, ankle, heel, and toe; and two additional markers placed on the middle of the left and right patella, respectively (as discussed above). The sampling rate was set to 250 Hz. The first and last instance of ground contact was detected by a force plate (AMTI, Optima HPS, Watertown, MA, USA) with a sampling rate of 2500 Hz synchronized with the Vicon system. For every participant and every trial, kinematic data from the first ground contact till take off were taken for further analysis.

To quantify the success of the knee alignment instruction, the landing phase was observed, being the crucial phase of ACL injury events. Instructions to move the knee over the foot axis aim at preventing inward rotation and medial displacement of the knee (valgus position). The maximal medial displacement of the knee in landing is expected to occur at the deepest squat position. Due to the higher reliability of the knee marker to identify knee position compared to the patella marker (which was added to aid the external focus condition), knee alignment was diagnosed by the trajectory of the marker at the femoral epicondylus lateralis in the medio-lateral direction between the first ground contact and the deepest squat position. This medio-lateral marker displacement was measured for each athlete and trial for the left and right leg, respectively, and averaged, creating the variable “knee-displacement”. Positive differences indicate a medial displacement of the knee relative to the foot (valgus position); negative differences indicate a lateral displacement. Knee-displacement was compared among uninstructed, internally focused, and externally focused trials. In addition, the medio-lateral knee position relative to the toe marker at the first ground contact was detected to identify potential differences in the initial position of the landing phase.

For the analysis of coordination patterns, the right scapula marker was omitted from the analysis. In the case of occluded markers, a custom gap-filling algorithm was applied [33,34]. One dataset had to be removed since the C7 marker was missing during the whole ground contact phase of all trials. Thus, the landing and jumping movements of 23 participants during the ground contact phase over three sets of five jumps were analyzed. Due to inter-person and between-trial differences in ground contact duration, the datasets were time normalized to 140 frames, which represented a typical trial duration. Marker position data were centered on the coordinates of the first ground contact of the right toe to compensate for different landing areas in different trials. A data matrix was constructed with participants and trials over time points (23 × 15 × 140) creating rows and marker coordinates (40 × 3) creating columns. A principal component analysis (PCA) was calculated from this input matrix using the PManalyzer [35], an open-source software application run in Matlab (Matlab R2018a, The MathWorks, Natick, MA, USA).

Principal components, constructed by weighted combinations of marker coordinates (principal movements (PMs) [26]), specify different movement patterns. Every single movement pattern (PM) represents a percentage of the whole movement (explained variance). If the coordination of a movement changes, a significant change in explained variance for the PMs will be observed. To address movement adaptations on an individual level, relative variances were calculated from the PCA scores [33]. This subject-specific relative variance (rVAR%) corresponds to the PCA eigenvalues, but is separately calculated for each athlete, trial, and PM. Only the rVAR% of PMs contributing more than 1% to the whole variance was considered in the current study.

### 2.4. Statistical Analysis

Outcome variables are the knee-displacement, as well as the subject-specific relative explained variance (rVAR%) of the considered PMs. To compare the outcome variables, 2-way repeated measures ANOVAs were conducted with the repetition factors “instruction” (uninstructed = CON, external focus = EF, and internal focus = IF) and “trials” (5 repetitions), for each PM separately. In case of the violation of the sphericity assumption (indicated by a significant Mauchly test), the degrees of freedom were corrected by the Greenhouse–Geisser procedure. Normality was checked using Shapiro–Wilk’s tests, and if non-normality was observed, Friedman tests for the repetition factor “trials” and “instruction” were conducted separately. In the case of significant results, Student’s t-tests for paired samples or Wilcoxon tests of set means were conducted.

To test whether the sequence of instruction (EF before IF or vice versa) or the focus preference (EFpref or IFboth) might affect the results, additional statistical analyses were conducted, which considered the grouping factors “sequence” or “preference”. None of these factors had an effect, and therefore, both factors were no longer considered.

All values are expressed as the mean ± standard deviation or 95% confidence interval. Statistical significance was set to α = 0.05. Multiple testing for the considered PMs was corrected using the Holm–Bonferroni procedure. Effect sizes are reported utilizing partial eta squared with effects lower than 0.01 considered as small, above 0.06 as moderate, and bigger than 0.14 as large [36]. In the case of Friedman’s test, Kendal’s W was used (<0.3 small, <0.5 moderate, >0.5 large) as the effect size and Cohen’s d (>0.2 small, >0.5 moderate, >0.8 large), as well as r (Z/sqrt(N)) for pairwise comparisons, respectively. All statistical calculations were completed in SPSS (IBM, Armonk, NY, USA, version 25).

## 3. Results

### 3.1. Knee Alignment

Instruction had a large effect on knee-displacement (F(1.5,33.9) = 11.68, *p* < 0.001, η_p_^2^ = 0.35), where the medial knee displacement was lowest with IF (Figure 1). Neither an effect of trials (task repetition), nor an interaction of instruction and trials were detected. Knee-displacement differed significantly between CON and EF, as well as CON and IF (t(22) = 4.13, *p <* 0.001, d = 0.86 and t(22) = 3.11, *p* = 0.006, d = 0.63, respectively).

The comparison of IF and EF revealed a more laterally displaced knee in the EF condition, but did not reach significance (t(22) = 1.90, *p* = 0.071, d = 0.40). Knee position at first ground contact changed between instructions (F(1.5,32) = 5.83, *p* = 0.013, η_p_^2^ = 0.21), but was approximately equal in the IF and EF condition, showing a straighter initial positioning of the knee than in CON.

### 3.2. Principal Components

The first four principal components (principal movements (PMs)) explained more than 96% of the variance in the landing-then-jumping movement, and every PM up to PM 4 individually explained at least 1% of the movement variation. Table 1 gives an overview of PM 1 to PM 4 with a short description and the averaged relative variance for all trials (Table 1).

The statistical analysis of the factors instruction (CON, EF, IF) and trials (five trials) revealed a significant effect of instruction in rVAR% of the first and third PM (F(1.4,31.2) = 6.32, *p* = 0.01, η_p_^2^ = 0.22.; χ^2^(14,23) = 34.4, *p* = 0.002, W = 0.11, respectively). There was no effect of trial, nor an interaction effect of trial and instruction. In PM 1, significant rVAR% changes were seen in IF compared to CON (t(22) = 2.85, *p* = 0.009, d = 0.59), but no difference in EF compared to CON (t(22) = 2.49, *p* = 0.021, not significant due to Holm–Bonferroni correction, d = 0.52) and IF compared to EF (*p* = 0.58). In PM 3, rVAR% of the IF and EF trials differed from the rVAR% of the CON trials (Z = 2.89, *p* = 0.004, r = 0.60, Z = 2.25, *p* = 0.024, r = 0.47, respectively), whereas IF and EF did not differ (Figure 2).

## 4. Discussion

This study compared the effect of internally vs. externally focused instructions on whole-body movement during a landing-then-jumping task. Knee alignment and movement pattern complexity were compared in trials of uninstructed (CON), externally focused (EF), and internally focused (IF) task execution.

The main findings of the study are that the instructions caused (i) a decrease in medial displacement of the knee, which has been associated with the risk of ACL injury, and (ii) a significant change in the contribution of the first and third movement component to the movement variation. Throughout all comparisons, no interaction of focus and trials was seen, demonstrating that there was no systematic change in how the instructions were implemented over the course of the testing protocol. Knee-over-foot alignment was quite accurate in the IF and the EF conditions with approximately equal medio-lateral knee position at initial ground contact. This does not support our first hypothesis, where we assumed that EF instructions would further enhance knee alignment compared to the IF or CON conditions. Both IF and EF instructions worked well for achieving desired movement form changes.

These results constitute good news for coaches and trainers: both instructions inducing an internal or an external focus of attention immediately and significantly changed movement execution despite running a well-trained task. Both IF and EF instructions clearly altered knee displacement towards the intended direction. Kristiansen et al., reported comparable results in free weight bench press strength training where both IF and EF instructions particularly changed muscle activity in addressed focus regions (barbell movement, pectoral muscle), but not in lower body muscles [37]. Both focus conditions provoked higher muscle activation compared to the baseline, underpinning the conscious control effect when aiming at parts of the movement execution [37,38,39]. Vidal et al., verified that internal focus instructions for the standing long jump (addressing knee extension) provoked more use of knee motion compared to unspecific external instruction of jumping (as near as possible towards a cone), provoking both better knee and ankle movement in jumping [40]. In our study, both types of instruction addressed one segment of the movement (knee alignment), revealing no difference in externally or internally focused instructions on task achievement. It might be that a more distal external focus could have helped to support spatial orientation. Haines et al., placed poles at shoulder width in front of the landing area and directed the external focus into pointing the knees towards the poles [41]. The valgus position at the initial ground contact, but not the peak valgus, differed between IF and EF conditions [41]. This change of knee positioning at the initial ground contact compared to CON was also observed in our study, but not when comparing IF and EF conditions. In addition, using poles as landmarks, visual information was available, enhancing the online control of the envisioned knee position. This is in contrast to the EF instruction in our study based on the imagination of patella trajectories in relation to the foot-marker position without visual control. The lack of visual information in our EF instruction may explain the lack of differences in movement form between our EF and IF conditions.

The second main result, the changes in the relative contributions of specific movement patterns (PMs) to the overall explained variance, depicts changes in the movement complexity. The effect was significant for the IF condition, showing a higher percentage in PM 1 and a lower one in PM 3 compared to the CON condition. Both PM 1 and PM 3 are whole-body movements, not specific knee movements (Table 1), showing that our knee-focused instructions affected the whole-body coordination. With the relative variance of PM 1 increasing and PM 3 decreasing in the IF condition, it can be concluded that the complexity of the whole-body coordination pattern is lowered (i.e., more variance is explained by less movement components). This may be interpreted as a “freezing” effect, a simplification of the coordination pattern, which is often associated with the early stages of learning [42,43]. Specifically, the freezing seems to manifest as a reduction in trunk bending and elbow movement (PM 3) in the IF and EF conditions compared to the CON condition. As a consequence, the main movement component, the impact absorption movement (PM 1), contributed more when instructions were given (this effect was significant for IF). Thus, our second hypothesis is partially confirmed: we observed a freezing effect in the IF condition compared to CON, as well as, but to a lesser degree (lower effect sizes), between EF and CON. We did not observe a significant difference in movement complexity between IF and EF in any of the PMs.

These results are in line with studies detecting tighter coupling of joint movements and reduction in movement variability for IF conditions [16,40,44] caused by conscious control [23,45]. In contrast to our expectations, the EF instructions also caused freezing of the degrees of freedom. This might refer to the fact that, in our study, the movement form was addressed, directing the athletes’ focus towards the movement trajectory. Monitoring one’s movement, e.g., with bodily information, has been said to interrupt automatized execution of the movement as a whole (known as the constraint action hypothesis) [1]. Hossner and Ehrlenspiel could verify that these impairments are only present in the focused point of the movement sequence, normally being an important nodal point of the movement technique (nodal point hypothesis) [39]. Knee alignment here could be defined as a nodal point or, better, as a nodal phase, where monitoring causes freezing and lowers the effectiveness of movement execution. It is important to note is that the current study tried to differentiate between externally or internally worded instructions on a nodal phase within a movement technique. Interestingly, in our study, this freezing effect tended towards smaller effect sizes in the EF condition than was seen in the IF condition, but without reaching significance in any direct comparisons.

Some limitations of our study have to be mentioned. The submaximal jump height in the different trials was not controlled since this was not the prevalent outcome. Hence, we cannot rule out that jump height differed between conditions; however, for the IF–EF comparison in particular, we observed non-significant results in movement form. Jumping height was not explicitly addressed to not interfere with landing instructions distracting the focus on knee alignment. Second, this study only refers to immediate adaptations of landing and jumping kinematics. Retention of an adapted landing technique could be observed [21], and a training of drop jump exercises could confirm transfer to side stepping [22], albeit that both findings rely on short observation periods. The persistence of EF effects after longer-lasting training or transfer to various non-trained conditions would still be of further interest in the context of injury prevention.

## 5. Conclusions

In summary, obvious valgus correction in landing was seen under EF and IF instructions, indicating a higher effect under the EF condition. EF, as well as IF instructions on knee alignment produced a lowering of movement complexity, or “freezing of degrees of freedom”, suggesting constrained and less automatized movement coordination compared to the no-instructions condition.

## Figures and Tables

**Figure 1 ijerph-19-10763-f001:**
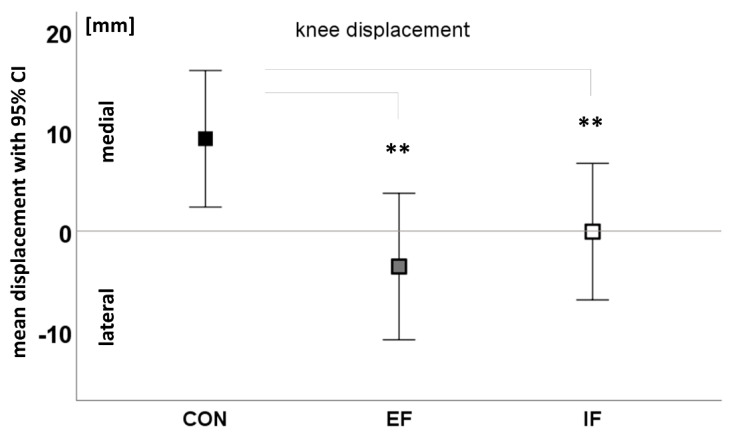
Main effect of focus on knee alignment (frontal displacement of knee-marker from first ground contact to deepest squat position) without instruction (CON) and with external focus (EF) and internal focus (IF) (** significant comparisons *p* < 0.01).

**Figure 2 ijerph-19-10763-f002:**
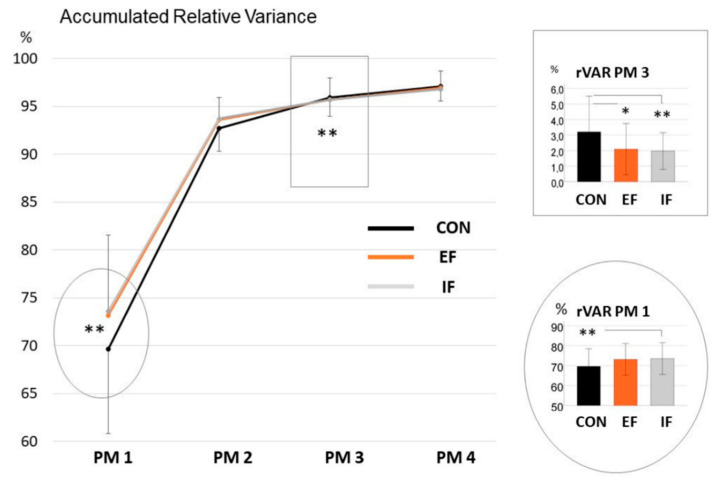
The large panel shows the accumulated subject-specific relative variance for the first four principal movements (PMs) (**left**) in the CON (without instruction), IF (internal focus), and EF (external focus) conditions over all trials. The small panels show the subject-specific variance for the first PM (**round panel**) and third PM (**square panel**) (** *p* < 0.01, * *p* < 0.05).

**Table 1 ijerph-19-10763-t001:** Principal movements (PMs) explaining more than 1% of the variation, mean explained variance with a short description, and visualization by athletes’ positions for the highest and lowest score of the selected PM.

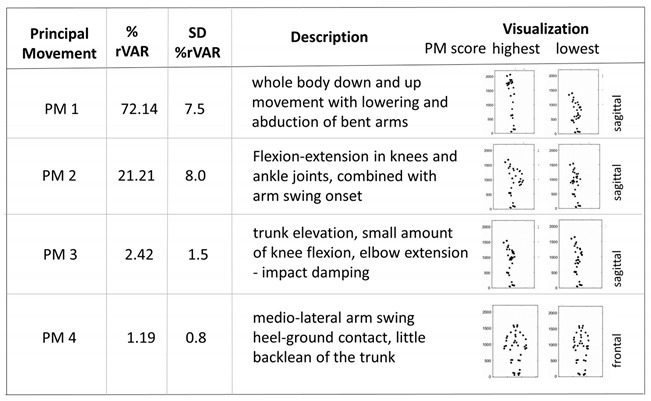

## Data Availability

The data presented in this study are available upon request from the corresponding author.

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
