# Peer review of "Intervention for Better Knee Alignment during Jump Landing: Is There an Effect of Internally vs. Externally Focused Instructions?"

_ijerph, 2022, doi:10.3390/ijerph191710763_

Round 1
Reviewer 1 Report
Werner et al. have investigated the effects on knee trajectory and whole-body movement complexity when addressing knee alignment using external and internal focused instructions. Knee alignment instructions led to significantly lower medial knee displacement compared to controls without instructions. These results are original and significant to the field, and they highlight the importance of instructions in the athletic performance of ski racers. This manuscript is extremely well-written and scientifically sound. It is refreshing to see that the authors have described their methodology in detail and highlighted potential shortcomings. All their claims are appropriately supported by their data and existing literature. Though the sample size is not large, I think the results are valid and signficantly add to the knowledge in the field. On the whole, I support publication of this manuscript and look forward to reading future work by this group.
Author Response
Response to Reviewer 1:
Thank you very much taking your time to review our paper! Thank you for giving such a positive feedback. We are working hard on our next publication.
Reviewer 2 Report
The aim of this paper is to compare the effect of an external versus an internal focus of attention in instructions on knee alignment when performing a land-then-jump task. Indeed, understanding the force's effect on body movements is important to prevent different injuries. There are some comments and concerns that should be addressed.
Page 1, lines 28-34 please cite these sentences appropriately.
Inclusion and exclusion criteria are not reported, please add them to the method section.
The data collection needs some clarifications, I suggest adding figures or pictures including the assessment process.
The discussion section is too long, I suggest making it shorter.
